# Predictive Computational Model for Damage Behavior of Metal-Matrix Composites Emphasizing the Effect of Particle Size and Volume Fraction

**DOI:** 10.3390/ma14092143

**Published:** 2021-04-23

**Authors:** Shaimaa I. Gad, Mohamed A. Attia, Mohamed A. Hassan, Ahmed G. El-Shafei

**Affiliations:** Department of Mechanical Design and Production Engineering, Faculty of Engineering, Zagazig University, Zagazig P.O. Box 44519, Egypt; sh.gaad@gmail.com (S.I.G.); mohdebada58@yahoo.com (M.A.H.); agelshafei@yahoo.com (A.G.E.-S.)

**Keywords:** particulate-reinforced metal matrix composites (PRMMCs), damage mechanisms, random microstructure-based model, volume fraction, particulate size, finite element method (FEM)

## Abstract

In this paper, an integrated numerical model is proposed to investigate the effects of particulate size and volume fraction on the deformation, damage, and failure behaviors of particulate-reinforced metal matrix composites (PRMMCs). In the framework of a random microstructure-based finite element modelling, the plastic deformation and ductile cracking of the matrix are, respectively, modelled using Johnson–Cook constitutive relation and Johnson–Cook ductile fracture model. The matrix-particle interface decohesion is simulated by employing the surface-based-cohesive zone method, while the particulate fracture is manipulated by the elastic–brittle cracking model, in which the damage evolution criterion depends on the fracture energy cracking criterion. A 2D nonlinear finite element model was developed using ABAQUS/Explicit commercial program for modelling and analyzing damage mechanisms of silicon carbide reinforced aluminum matrix composites. The predicted results have shown a good agreement with the experimental data in the forms of true stress–strain curves and failure shape. Unlike the existing models, the influence of the volume fraction and size of SiC particles on the deformation, damage mechanism, failure consequences, and stress–strain curve of A359/SiC particulate composites is investigated accounting for the different possible modes of failure simultaneously.

## 1. Introduction

Particle-reinforced metal matrix composites (PRMMCs) combine two or more constituents to tailor their best properties like strength and modulus of the reinforcement, and ductility and toughness of the matrix. Metal matrix composites (MMCs) reinforced with micro/nanoparticles are widely used in several engineering application, i.e., aerospace, aviation, and arms structural components, due to their high specific strength, high fracture toughness, high thermal conductivity, better abrasion resistance, enhanced corrosion resistance, and other features [1,2]. Among PRMMCs, aluminum matrix composites (AMCs) are extensively used owing to their good mechanical performance, which depends on the content, shape, size, and dispersion of the reinforcing particles [3,4].

In composite materials, damage can initiate and propagate macro-cracks leading to the failure of their structures. In PRMMCs, different damage mechanisms can occur such as matrix yielding, formation of voids around the reinforcing particles owing to the weak interfacial bonding or at cracked matrix adjacent to the particle, and the cracked particles [2]. In the analysis of damage behavior of PRMMCs, experimental testing may not predict the onset of each failure mode or the required applied strain to initiate this mode. Additionally, the estimation of the effective plastic strain and the equivalent stress during the deformation of the material is difficult using the experiments. Therefore, experimental results cannot completely give insights into the damage evolution and consequences of the possible failure modes. Therefore, several computational finite element (FE) odels have been proposed to simulate the damage mechanisms of PRMMCs based on macro-scale and micro-scale levels. In the macro scale level approach, the multiphase PRMMC is replaced by an equivalent single-phase isotropic homogeneous material ignoring different defects of the real composites [3]. This approach leads to unsatisfactory results and can only provide the macro elastoplastic behavior of the composite. The micro-scale level approach, which is also known as the microstructure-based numerical model, can be applied via several techniques: (i) employing image processing tools to model the real microstructure taken from SEM image [5], (ii) Assuming an idealized regular arrangement of the particles, and thus, the real microstructure of PRMMCs cannot be captured [6], (iii) Representative volume element (RVE) model was adopted by some authors for saving the computational time [7,8,9], and (iv) Random distribution-based modelling of reinforcing particulate, which reflects the real microstructure of the composite. However, except for the first technique, the others cannot well simulate the particle clustering and voids which may exist in PRMMCs [9,10].

Failure of PRMMCs is predominantly by the nucleation and growth of voids occurred by the deboning between matrix and particles and the particle fracture [11]. Mechanism of particle–matrix interfacial debonding has been extensively investigated via cohesive zone models (CZMs) [12]. The carbon nanotube (CNT)/matrix interface in a well-arrayed CNTs array composite was characterized employing a micromechanics-based bilinear cohesive FE model [13]. The overall properties of nanocomposites are remarkably impacted by the interface properties. A CZM based on FE was developed to evaluate the interface debonding in PRMMCs [2]. It was noticed that no particle–matrix debonding occurs if the cohesive energy is larger than its critical value at the interface. Elastoplastic damage behavior based on the particle–matrix interfacial bonding of Al-5%Al_2_O_3_ nanocomposites was investigated using RVE and the cohesive element method [14]. It was reported that the stress–strain behavior was considerably affected by the interface debonding. Neither the matrix cracking nor the particle fracture is considered in this model, which affects the obtained results.

A CZM using RVE was presented to examine the impact of particle–matrix interface damage on the performance of PRMMCs [15]. Two models with spherical and cubic particles with identical contents and distributions were studied. The composite strength is reduced by the interface damage, especially with cubic particles. CZM is employed to model the interfacial debonding for investigating the influence of debonding at the interface between Al and SiC [16]. The actual microstructure and idealized ellipsoids of SiC powders were studied. It was reported that simplifying the particle morphological characteristics may lead to inaccurate analysis. On the basis on micromechanics, an analytical model was proposed to examine the influence of matrix cracking and interface debonding on the stress–strain curves of PRMMCs [17].

A 2D CZM was proposed to study the mechanical performance of PRMMCs accounting for the fracture of matrix and particles [8,9]. It was concluded that the interface strength and the particle arrangement have considerable influences on the strength and failure strain of PRMMCs. An RVE model was proposed to investigate fracture behaviors in Al6061/SiC composites using CZM for interface decohesion and Griffith criterion for the particle fracture [18]. A high interfacial Al/SiC bonding was noticed and the interfacial debonding occurred during the crack growth from the particle to the interface. Failure mechanisms in MMCs under tensile loading condition was investigated using a microstructure-based model [19]. The arrangement of particles had a negligible influence on the predicted stress–strain behavior and considerably influenced the position and shape of the formed cracks and their growth. The particle cracking starts at its sharp corners without any noticeable interfacial debonding. Recently, the damage mechanisms of PRMMCs reinforced with different shapes of particles employing FE model based on random distribution of particles. The effect of reinforcing circular, hexagonal, square, and triangular particles and their combinations is investigated. The circular-shaped particle was found to be the best choice to be reinforced in PRMMCs because it has the maximum UTS, and the maximum required applied load to reach its value [20]. 

The effect of the particle size on the size-dependent plastic behavior of PRMMCs has been captured through the continuum theory of mechanism-based strain gradient plasticity. Within the conventional theory of mechanism-based strain gradient plasticity, the particle size effects in composites have been explored considering the particle/matrix interface decohesion [21]. A FE model including the effects of load transfer, grain refinement, thermal residual stress/strain, plastic strain gradient, and matrix damage was developed to assess the tensile loading behavior of the 2009Al/15%SiC composite [10]. Employing the Taylor-based nonlocal theory of plasticity besides the cohesive zone model for interfacial debonding, the particle size-dependent behavior of PRMMCs is investigated [22]. In the context of the continuum theory of stress gradient plasticity, the plastic behavior of PRMMCs was studied at different particle sizes without including the interface decohesion or the particle fracture [23]. The strain gradient continuum plasticity in the context of the theoretical, experimental, and numerical investigations is comprehensively reviewed in [24]. 

From the above literature survey, it is seen that the previous studies are limited to the investigation of the effect of volume fraction or size of reinforcing particles on damage behavior of PRMMCs without accounting for the simultaneous presence of cracking of the elastoplastic matrix, debonding at the matrix–particle interface, and the fracture of reinforcing particles. This study aims to explore the influence of volume fraction and size of SiC particles on the damage mechanism, deformation, fracture, and mechanical performance of A359/SiC composites considering all the three possible failure modes. Numerical simulations are performed using an integrated 2D microstructure-based FE model in conjunction with image processing. The model considers the random distribution of particles to achieve the closest distribution to the real microstructure. The elastoplastic behavior and cracking of matrix, decohesion at the particle–matrix interface, and fracture of particles are modelled using Johnson–Cook plasticity and extended damage models, cohesive zone surface method, and elastic–brittle cracking model, respectively. ABAQUS/explicit FE software combined with Digimat FE commercial code is utilized to perform simulations. The influence of the strain gradient is beyond the main purpose of the present study. 

## 2. Modelling of Failure Mechanisms

### 2.1. Cracking of the Elastoplastic Matrix

In the present study, the plastic deformation of the matrix is described using Johnson–Cook constitutive relation [25]
(1)$$\sigma =\left(A+B\varepsilon^{n}\right),$$
where ε, n, A and B represent the equivalent plastic strain, the strain hardening coefficient, yield strength and strain hardening constant, respectively. The ductile damage of the matrix is modelled adopting the Johnson–Cook extended model [20,26],
(2)$$\varepsilon_{f}=D_{1}+D_{2}\exp\left[D_{3}\left(\sigma_{m}/\sigma_{eq}\right)\right],$$
where $\varepsilon_{f}$ and $\sigma_{m}$ are the equivalent fracture strain and the mean stress, respectively. The fracture constants $D_{1}\;,\;D_{2\;}$ and $D_{3\;}$ are material-dependent. The stress in the damaged state of the material is given by $\sigma_{D}=\left(1-D\right)\sigma_{eq}$, where D  represents the damage parameter,
(3)$$D=\sum \left(\Delta \varepsilon /\varepsilon_{f}\right),$$
where Δε is the incremental equivalent plastic strain. It is noted that 0≤D<1  and D=1.0 means that the element is completely failed.

For the matrix constituent (A359 alloy), the values of modulus of elasticity, yield strength, ultimate tensile strength, Poisson’s ratio, and density are, respectively, 73.9 GPa, 127 MPa, 182 MPa, 0.33, and 2670 kg/cm^3^ [19]. In Equation (1), A, B and n are 83.1 MPa, 337.5 MPa, 0.3545 [19] and in Equation (2), $D_{1}\;,\;D_{2\;}$ and $D_{3}$ are 0.0044, 0.2368, and −2.775, respectively [20].

### 2.2. Particle–Matrix Interface Decohesion

The matrix–particle interface is modelled adopting a bilinear CZM that allows for normal and tangential separations [12]. The quadratic nominal stress criterion is adopted for detecting the initiation of damage, such that [27]
(4)$${\left\{\frac{\langle T_{n}\rangle }{T_{n}^{o}}\right\}}^{2}+{\left\{\frac{T_{s}}{T_{s}^{o}}\right\}}^{2}+{\left\{\frac{T_{t}}{T_{t}^{o}}\right\}}^{2}=1,$$
where $T_{n}$, $T_{s}$, and $T_{t}$ are, respectively, the normal and the two shear tractions, $T_{n}^{o}$ denotes the interfacial normal strength and $T_{s}^{o}\;$, and $T_{t}^{o}$ denote the two interfacial shear strengths. The Macaulay bracket 〈 〉 implies that damage onset is not affected by the compressive normal stress, i.e., $\langle T_{n}\rangle =T_{n}$ if $T_{n}>0$, else $\langle T_{n}\rangle =0$. The stress components are influenced by the damage as follows [28]:(5)$$T_{n}=\{\begin{array}{l} \left(1-D\right){\bar{T}}_{n},\;\;\;\;\;\;\;\;\;{\bar{T}}_{n}\geq 0\\ {\bar{T}}_{n},\;\;\;\;\;\;\;\;\;\;\;\;\;\;\;\;\;\;\;\;\;\;\;{\bar{T}}_{n}<0 \end{array},\;\;T_{s}=\left(1-D\right){\bar{T}}_{s},\;\mathrm{and}\;\;T_{t}=\left(1-D\right){\bar{T}}_{t},$$

The overbar refers to the corresponding stress components at the current strain before the onset of damage. The scalar damage variable D representing the overall damage at a contact point is given by
(6)$$D=\frac{\Delta_{m}^{f}\;\left(\Delta_{m}^{max}-\Delta_{m}^{o}\right)}{\Delta_{m}^{max}\;\left(\Delta_{m}^{f}-\Delta_{m}^{o}\right)},\;\;\;\;\Delta_{m}^{o}<\Delta_{m}^{max}\;\leq \Delta_{m}^{f},$$
where $\Delta_{m}^{o}$ and $\Delta_{m}^{f}$ are the effective separations at the damage initiation and complete failure, respectively,$\;\Delta_{m}^{f}=2G_{C}/T_{n}^{0}$ in which $G_{C}$ is the fracture energy.$\;\Delta_{m\;}^{max}$ is the maximum effective displacement during the loading history. The effective separation $\delta_{m}^{\;}$ can be expressed as [28]
(7)$$\delta_{m}^{\;}=\sqrt{\langle \delta_{n}^{\;}\rangle {}^{2}+\delta_{s}^{2\;}+\delta_{t}^{2\;}\;\;},$$
in which $\delta_{n}^{\;}$ is the normal separation and $\delta_{s}^{\;}$ and $\delta_{t}^{\;}$ are the two shear separations.

For A359/SiC interface, the fracture energy $G_{C}$ is 50 J/m^2^ [29], elastic modulus and shear moduli are 180.6 GPa and 76.6 GPa, respectively [30], and the cohesive strengths in both normal and shear directions are 372 MPa [31].

### 2.3. Fracture of Reinforcing Particles

Within the context of smeared crack concept, the fixed-orthogonal model is employed to model the elastic–brittle fracture of the reinforcing particulates. Once, the crack is initiated, it propagates perpendicularly to the maximum tensile stress [32]. Although the crack detection is only based on Mode I, the post-cracked behavior includes Mode II as well as Mode I. At the complete loss of integrity, the crack normal displacement $u_{no}$ can be defined as
(8)$$u_{no}=2G_{f}^{I}/\;\sigma_{tu}^{I},$$
where $G_{f}^{I}\;$ represents the energy needed for opening a unit area of a crack in Mode I and $\;\sigma_{tu}^{I}$ denotes the failure stress. The shear moduli of the cracked ($G_{S}$) and uncracked (G) particles are related by
(9)$$\;G_{S}=G{\left(1-\frac{\varepsilon_{n}^{c}}{\varepsilon_{nf}^{c}}\right)}^{p},$$
where $\varepsilon_{n}^{c}$ and $\varepsilon_{nf}^{c}$ are, respectively, the crack opening normal strain and the stress-free crack normal strain and p is a material parameter controlling the shear retention. For SiC particles, the modulus of elasticity, ultimate tensile strength, Poisson’s ratio, and mass density are, respectively, 410 GPa, 400 MPa, 0.14, and 3210 kg/cm^3^, p = 2, and $\varepsilon_{nf}^{c}\;$= 0.2 [33].

## 3. The Finite Element Model

The mathematical formulations of different failure modes described in Section 2 are implemented into a 2D nonlinear FE model using ABAQUS/Explicit software (version 2020 Providence, RI, Dassault Systèmes Simulia Corp, Johnston, RI, USA) package. Utilizing the image processing technique, Digimat FE commercial code is employed to develop the random arrangement of particles in the matrix [34]. Figure 1 illustrates a 2D FE model of the tensile test of A356/SiC composite, in which the lower edge is constrained in *y*-direction, except the central node is fixed and the upper edge is exposed to a uniform displacement $u_{y}$ of 10 µm. In the present numerical simulations, the 2D FE model consists of 4-node bilinear, reduced integration with hourglass control elements (CPE4R) and linear constant strain triangular elements (CPE3). For accurate predictions of the damage behavior and failure modes, a fine FE mesh is applied to the entire microstructure, as shown in Figure 1. To this end, the estimated engineering stress $\sigma_{e}$ and strain $\varepsilon_{e}$ are converted to their corresponding true stress σ and true strain ε, such that $\sigma =\sigma_{e}\left(1+\varepsilon_{e}\right)$ and $\varepsilon =\ln\left(1+\varepsilon_{e}\right)$. Additionally, the ultimate tensile strength (UTS) is defined as the maximum predicted value of stress and the failure strain $\left(\varepsilon_{f}\right)$ is recorded when a complete fracture of the microstructure occurs [35].

Convergence of the present FE model is checked by carrying out the simulations using four different FE meshes with 2D plane strain elements of sizes 0.35, 0.7, 1.0, and 1.4 µm. The predicted true stress–strain curves for A359/SiC composite with a particle size (PS) of 10 µm and volume fraction (VF) of 10%, shown in Figure 1, are plotted in Figure 2. The representative model in Figure 1 has a length and a width of 240 and 120 µm, respectively. Table 1 provides the extracted values of the ultimate tensile strength and the failure strain of the specimen. It is depicted that the present simulations give stable results with mesh sizes of 0.35 µm and 0.7 µm. Accounting for the accuracy of results and the computational cost, the FE mesh of size 0.7 µm is selected for all forthcoming simulations.

To verify the present computational FE model, consider the A359/SiC composite with the microstructure shown in Figure 3a. The predicted true stress–strain curve of this composite under tensile loading of $u_{y}$ of 10 µm and those experimentally and numerically reported in [19] are illustrated in Figure 3b. It is depicted that present predictions are in accordance with the experiment. The present value of the modulus of elasticity (99.8 GPa) is very close to its numerical (102.6 GPa) and experimental (98.6 GPa) predictions [19]. In addition, the present model predicts an ultimate tensile strength (UTS) of 200.23 MPa, which is in good agreement with its experimental value (210 MPa) [19]. The numerical value of UTS predicted in [19] was 171.69 MPa which is smaller than that of the pure A359 alloy (182 MPa).

## 4. Results

The developed integrated 2D computational FE model is employed to explore the impacts of volume fraction and size of SiC particles on the damage behavior, consequences of failure modes, and the required load for the onset of each failure mode in A359/SiC composite accounting for the mutual influences of various possible damage modes. The material parameters of the A359 matrix, SiC particles, and the interface between SiC and A359, presented in Section 2, are implemented. The reinforcing particles take an approximately spherical shape [14]. Employing the integrated damage model, the impact of VF and PS on the stress–strain curve and the mechanical properties such as the modulus of elasticity, yield strength, ultimate tensile strength, and failure strain are quantitively predicted, in addition to the distributions of the equivalent the von Mises stress and effective plastic strain.

To extract a clear exploration of the influence of VF and PS of the SiC particles on the damage mechanism and failure modes of A359/SiC composite, all parameters controlling the particulate effect are kept constant while changing its VF or PS. The A359/SiC specimen is exposed to a prescribed normal displacement of 10 µm at the top edge of the model, as illustrated in Figure 1.

### 4.1. Effect of SiC Particles Volume Fraction (VF)

The effect of VF of SiC particles is investigated by considering different microstructure including different VFs, i.e., 2%, 5%, 10%, 15%, and 20%, as shown in Figure 4. The particles are randomly distributed within the matrix, such that keeping the particles size constant at 10 µm, the positions of SiC particles at 2% VF are kept fixed for 5, 10, 15, and 20% VFs and those at 5% VF are kept fixed for 10, 15, and 20% VFs and so on.

Figure 5 depicts the simulation results of the true stress–strain curves at different VFs. The influence of VF on the modulus of elasticity, yield strength, ultimate tensile strength, and failure strain is provided in Table 2. The damage mechanisms and consequences of failure including the onset of each failure mode are presented in Figure 6, Figure 7, Figure 8, Figure 9 and Figure 10 at 2%, 5%, 10%, 15% and 20% VFs of SiC particles, respectively. The applied load (true strain) needed for initiation of each mode of failure as well as reaching the UTS and the associated true stress (MPa) are recorded. The effect of VF of SiC particles on the distributions of the effective plastic strain within the matrix material at the onset of matrix cracking and the complete failure are depicted in Figure 11.

### 4.2. Effect of SiC Particles Size (PS)

The influence of the SiC particle size (PS) on the deformation and damage behaviors of A359/SiC composite is investigated accounting for different microstructures including different PS of SiC particles, as demonstrated in Figure 12. In these microstructures, the positions of SiC particles within the matrix are randomly distributed keeping a constant 10% VF. 

Figure 13 depicts the predicted true stress–strain curves of the composites with different sizes of SiC particles and Table 3 provides the influence of SiC size on the mechanical properties of A359/SiC composite. The predicted damage behavior, consequences of failure, and the required load for the first occurrence of each damage mode in A359/SiC composite are presented Figure 14, Figure 15, Figure 16 and Figure 17 at a size of SiC particles of 2, 5, 15, and 20 µm, respectively. The effect of SiC size on the effective plastic strain within the matrix is demonstrated in Figure 18 at the onset of matrix cracking and complete failure.

For the sake of illustration, Figure 19 summarizes the applied strain needed for the beginning of possible failure modes and the complete failure of the composite at several values of VF and PS of SiC particles.

## 5. Discussion

### 5.1. Discussion of the Effect of SiC Particles Volume Fraction

From Figure 5, it is depicted that increasing VF of SiC increases the stress level on the true stress–strain curve in the undamaged state. Increasing VF above 10% causes some stress fluctuations during the evolution of damage. These fluctuations are noticeably observed in the composites which suffer more fracture in particles. Before the damage initiation, the strain hardening of A359 leads to increasing the stress, while the crack propagation reduces the stress on the true stress–strain curves. Based on the recorded mechanical properties in Table 2, it is observed that the higher modulus of elasticity of the composite is owing to the inclusion of hard SiC particles in the soft A359 matrix, which is in accordance with [36,37]. When the composite is loaded, the matrix strongly constrains the particles because of the strong bond between them, and thus, higher stress is needed to cause the same deformation compared to the unreinforced matrix. Compared to A359 alloy, reinforcement with SiC of 5% and 10% VFs raises the modulus of elasticity by 11.5% and 35.7%, respectively. Increasing VF of SiC increases the predicted yield strength and UTS and reduces the failure strain. This enhancement in the tensile strength is attributed to the improvement of the composite modulus by increasing VF. At high VF of SiC, a low failure strain means an easy failure of the composite. The rate of change in the recorded mechanical properties is reduced as the VF of SiC varies beyond 15%. However, it can be concluded that at the same applied strain, although the stress for the composites with high VF is greater than that with low VF, the composite with low VF continues sustaining load with improved strength and ductility properties. 

Considering the influence of SiC VF on the damage evolution, it is demonstrated from Figure 6, Figure 7, Figure 8, Figure 9 and Figure 10 that no decohesion is observed along the particle/matric interface, which is owing to the strong bond between the matrix and particles at all the studied values of VF. The damage mechanism starts with matrix cracking at low VF up to 10% and with a particle fracture at 15% and 20% VF. In other words, increasing VF raises the possibility of particle fracture, which is due to the modulus of elasticity increasing by the increasing VF, and consequently, the stress level becomes higher. Additionally, the particles exhibit no fracture at 5% VF, which may be owing to the random arrangement of particles in the matrix, as depicted in Figure 7. The obtained results show that the initiation of matrix cracking occurs just before the stress level approaches UTS, regardless of the VF of SiC particles. Increasing VF of SiC particles reduces the true strain needed to the onset of matrix cracking and in accordance increases the corresponding stress. The true strain corresponding to UTS decreases as VF increases. When the stress level exceeds UTS, the matrix is highly deformed and in turn, the matrix crack propagates at 45° along the applied displacement. Therefore, the stresses are considerably released on the crack sides and new matrix cracks are detected. These microcracks extend through the composite and grow until complete separation occurs. As mentioned before, the strain required for complete failure (failure strain) is noticeably reduced by increasing the VF of SiC particles. In addition, von Mises stress in the composite increases as the particle content increases due to the same reason of enhancing the modulus of elasticity.

The distribution and maximum equivalent plastic strain are considerably influenced by varying VF of SiC, as illustrated in Figure 11. By varying VF from 2% to 10% and from 10% to 20%, the maximum plastic strain is reduced by 18.1% and 55%, respectively, at the onset of matrix cracking, and is increased by 7.4% and 24.6%, respectively, at the complete failure of the composite. It is also observed that for a given applied strain, the equivalent plastic strain increases as VF of SiC increases.

### 5.2. Discussion of the Effect of SiC Particles Size 

The trends of the stress–strain curves as well as the mechanical properties are considerably influenced by PS, as depicted in Figure 12. The smaller size of SiC makes the composite exhibits high stress levels, and therefore, the flow stress and work hardening increase. Such response is owing to that reduction of SiC size strengthens the bonding at matrix–particle interfaces, which in turn raises the total load transferred to the reinforcing particles. It is revealed from Table 3 that the predicted modulus of elasticity is not influenced by varying PS. On the contrary, the ultimate tensile strength, yield strength, and fracture strain are decreased by 7.6%, 7.5%, and 4.4%, respectively, as PS varies from 2 µm to 10 µm, and by 10.9%, 1.1%, and 24.4%, respectively, as PS varies from 10 µm to 20 µm.

It is demonstrated from Figure 8 and Figure 14, Figure 15, Figure 16, Figure 17 that the composites with different PSs exhibit matrix cracking and particle fracture damage modes, whereas no decohesion is detected at the matrix–particle interface. The consequences of damage in A359/SiC composite are remarkably impacted by PS of SiC particles. Although the composites with 2 µm and 5 µm SiC follow the same damage behavior, the needed applied strain for starting of the same damage mode is different. As failure starts by the particle fracture, the stress concentrates at the matrix side near the fractured particle until the stress level reaches the tensile strength. The location of the first fractured particle is significantly influenced by PS. As the matrix plastic strain approaches its critical value, the matrix failure initiates. The matrix cracking grows, and new matrix cracks appear, extend through the composite at 45° to the applied displacement direction leading to the complete separation of the specimen. A similar trend of damage mechanism is noticed for the composite reinforced by PS of 20 µm, except that the matrix cracking occurs at an early loading stage before reaching UTS, as demonstrated in Figure 17. On the other hand, for the composite with PS of 10 µm and 15 µm, the stress concentrates at the matrix near the particles, as shown in Figure 8 and Figure 16, respectively. Increasing the applied strain, the stress in particles increases and the fracture of particles appears. The matrix cracking propagates resulting in a large crack until the specimen completely fails. Although the matrix cracking starts from the same locations for both particle sizes of 10 µm and 15 µm, the particle fracture starts in different locations, and thus, the cross-section images of the fractured specimens are different. Additionally, increasing PS significantly reduces the failure strain and the maximum von Mises stress. As the stress level in the composite reaches its UTS, varying PS from 2 µm to 5, 10, 15, and 20 µm results in a reduction in the predicted peak von Mises stress in the composite by 11.3%, 20%, 36.7%, and 62.7%, respectively.

It is noticeable from Figure 18 that changing PS from 2 to 20 µm reduces the maximum plastic strain at the onset of matrix cracking and complete failure by 40.3% and 36.9%, respectively. Whereas, as PS changes from 2 to 10 µm, the maximum plastic strain is increased by 32.9% at the onset of matrix cracking and is decreased by 6.9% at the complete failure.

It is observed from Figure 19 that both VF and PS of SiC particles considerably influence the needed strain for onset of different damage modes of A359/SiC composite. The required true strain for reaching the ultimate tensile strength, initiation of matrix cracking, the onset of particle fracture, and the complete failure are, respectively, reduced by 15.7%, 13.7%, 13.4%, 9.6% when VF of SiC changes from 2% to 10% and by 42.4%, 35.7%, 57.6%, 28.3% when VF changes from 10% to 20%. As PS of SiC increase from 2 µm to 20 µm, the required true strain for the initiation of cracking in matrix and particle fracture, approaching the ultimate strength, and the occurrence of complete separation are reduced by 54%, 63.5%, 20.1%, and 27.8%, respectively.

## 6. Conclusions

In this paper, an integrated numerical FE model is proposed for investigating the effects of two design parameters: volume fraction and size of reinforcing particles on the damage behavior of PRMMCs. To the best of our knowledge, this is the first study of the influences of the volume fraction and size of SiC particles on the damage mechanism, damage evolution, consequences of failure, and the mechanical performance of A359/SiC composites considering the simultaneous contributions of the different possible damage modes. In the framework of random microstructure-FE analysis, Johnson–Cook plasticity and extended damage models are employed to simulate the elastoplastic behavior and cracking of the A359 matrix material, respectively. The elastic brittle cracking model is adopted to capture the fracture of SiC particles. Surface-based-CZM is employed to model the A359–SiC interface decohesion. Implementing the proposed model into ABAQUS/Explicit software in parallel with Digimat software, the effects of VF and PS of SiC particles on the damage mechanism, consequences of failure, the required load for the onset of each failure mode, and mechanical properties of A359/SiC composite under tensile loading are comprehensively predicted and discussed. For the analysis of PRMMCs, the main conclusions are as follows:Increasing VF of SiC from 2% to 20% increases the modulus of elasticity, yield strength, and tensile strength and decreases the ductility of A359/SiC composite, by about 31.7%, 33.7%, 19.5%, and 35.2%, respectively. Whereas increasing the PS of SiC particles from 2 µm to 20 µm reduces the yield strength, ultimate tensile strength, and failure strain by, respectively, 8.6%, 17.7%, and 27.7% without changing the modulus of elasticity.The probability of fracture of SiC particles increases by rising its VF, whereas the damage mechanism starts with matrix cracking at low VF. Dependent on the PS and distribution of SiC particles, the damage mechanism of A359/SiC composite may be started by matrix cracking or particle fracture. No interfacial debonding at SiC–A359 interface appears as the stress concentration level remains below the interfacial strength.Under tensile loading, as VF SiC particles increases from 2% to 20%, the true strains needed for the onset of matrix cracking, particle fracture, complete failure, and reaching the ultimate tensile strength in A359/SiC composite decrease by, respectively, 44.4%, 63.3%, 35.1%, and 51.5%. Increasing the PS of SiC particles, the required strain for starting the matrix cracking, particle fracture, occurrence of complete failure, and approaching the ultimate tensile strength in A359/SiC composite reduce by 54%, 63.5%, 27.7%, and 20.1%, respectively.The contours and peak values of von-Mises stress in the A359/SiC composite and the effective plastic strain in the matrix are sensitive to VF and PS of SiC. Increasing VF of SiC particles from 2% to 20% increases the maximum effective plastic strain in the matrix A359 by about 20.5% and 33.9% at the onset of matrix cracking and the complete failure of the composite, respectively. On the other hand, the rising PS of SiC particles shows a decrease in the maximum effective plastic strain in the matrix A359 by 40.3% and 36.9% at, respectively, the matrix cracking initiation and the complete failure of the composite.

The developed FE model and the obtained results could be useful for optimization of PRMMCs accounting for the size and volume fraction of the reinforcing particles as design parameters.

## Figures and Tables

**Figure 1 materials-14-02143-f001:**
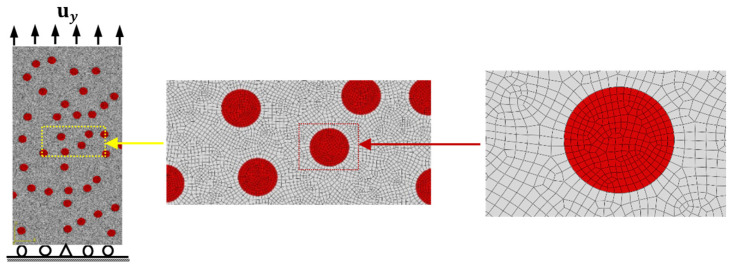
The FE model of A356/SiC specimen with loading and boundary conditions.

**Figure 2 materials-14-02143-f002:**
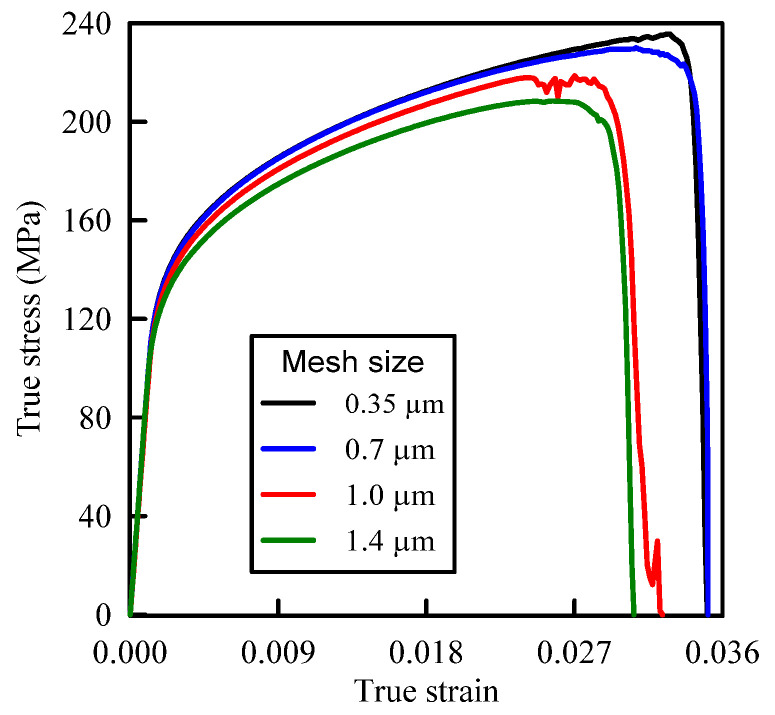
Tensile true stress–strain curves for A359/SiC composite using different FE mesh sizes (PS = 10 µm, 10% volume fraction (VF)).

**Figure 3 materials-14-02143-f003:**
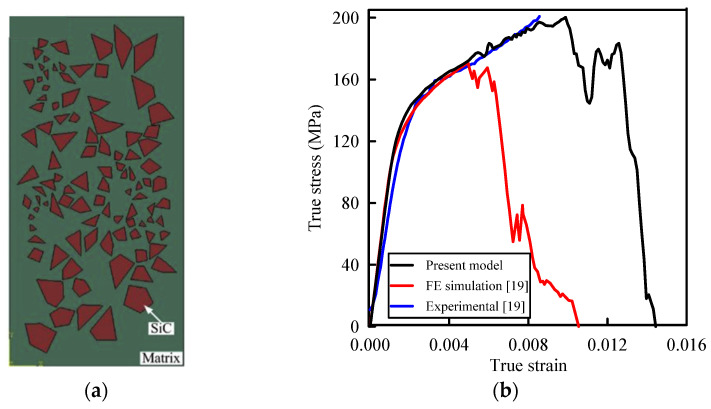
(**a**) Microstructure of A359/SiC composite (adapted with permission from ref. [19], Copyright 2016 Springer); (**b**) True stress–strain curves of A359/SiC composite.

**Figure 4 materials-14-02143-f004:**
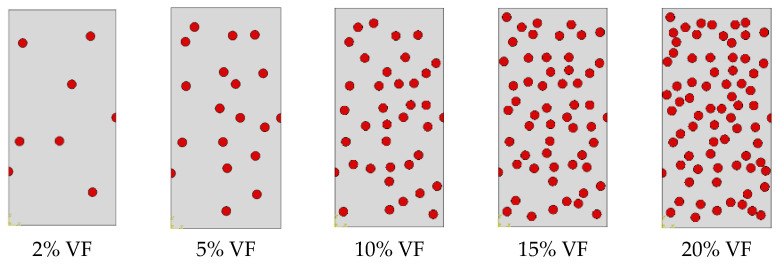
Representation of microstructure of A359 alloy matrix and SiC particles with various VFs (PS = 10 µm).

**Figure 5 materials-14-02143-f005:**
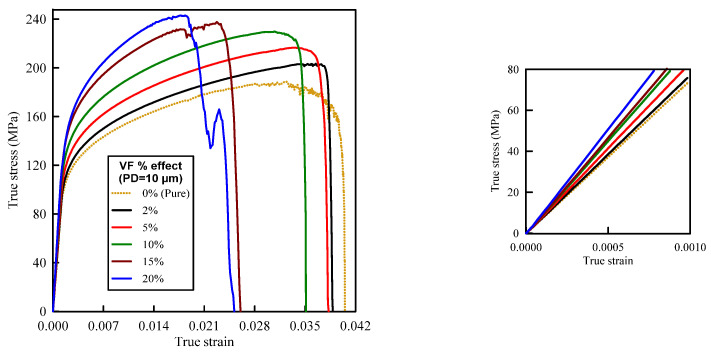
Effect of SiC particles volume fraction on the true stress–strain curve of A359/SiC composite (particle size (PS) = 10 µm).

**Figure 6 materials-14-02143-f006:**
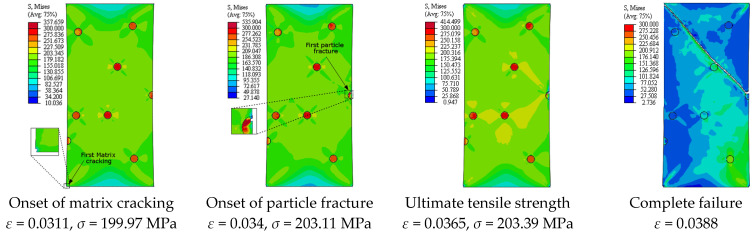
Failure propagation and von Mises stress distribution of A359/SiC composite at 2% VF (PS = 10 μm).

**Figure 7 materials-14-02143-f007:**
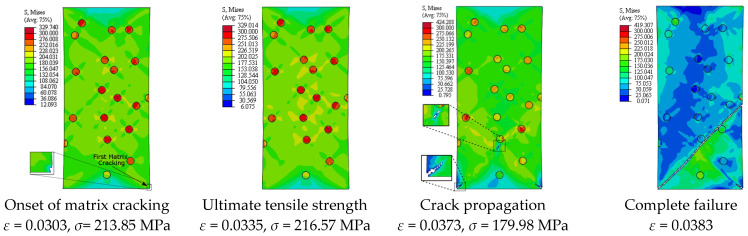
Failure propagation and von Mises stress distribution of A359/SiC composite at 5% VF (PS = 10 μm).

**Figure 8 materials-14-02143-f008:**
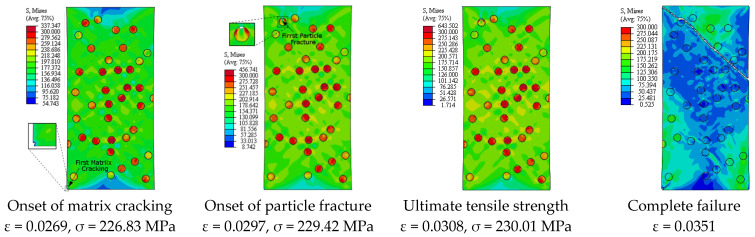
Failure propagation and von Mises stress distribution of A359/SiC composite at 10% VF (PS = 10 μm).

**Figure 9 materials-14-02143-f009:**
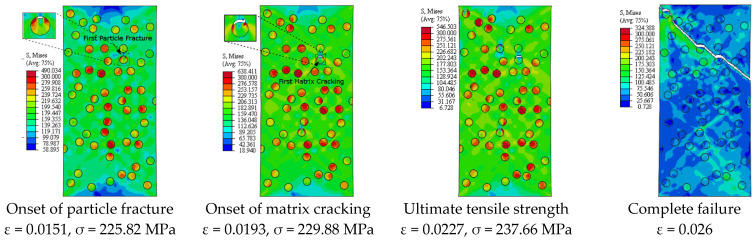
Failure propagation and von Mises stress distribution of A359/SiC composite at 15% VF (PS = 10 μm).

**Figure 10 materials-14-02143-f010:**
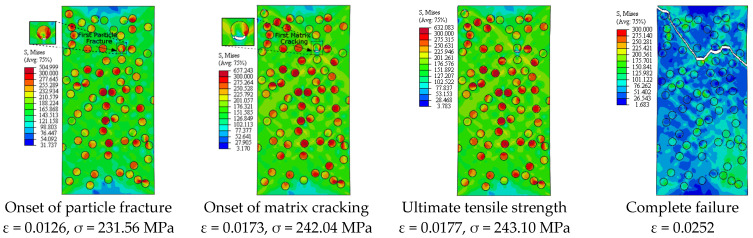
Failure propagation and von Mises stress distribution of A359/SiC composite at 20% VF (PS = 10 μm).

**Figure 11 materials-14-02143-f011:**
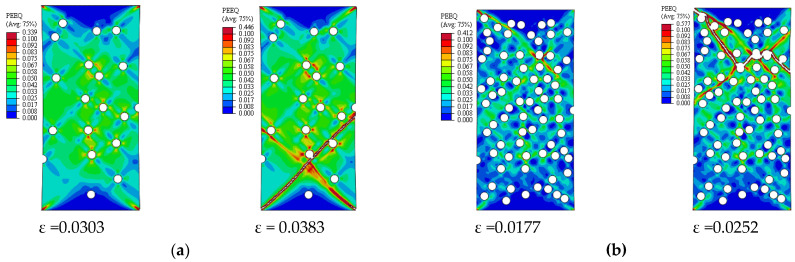
Effective plastic strain in the matrix of A359/SiC composite (PS = 10 µm): (**a**) 5% VF; (**b**) 20% VF.

**Figure 12 materials-14-02143-f012:**

Representation of microstructure of A359 alloy matrix and SiC particles with different sizes (10% VF).

**Figure 13 materials-14-02143-f013:**
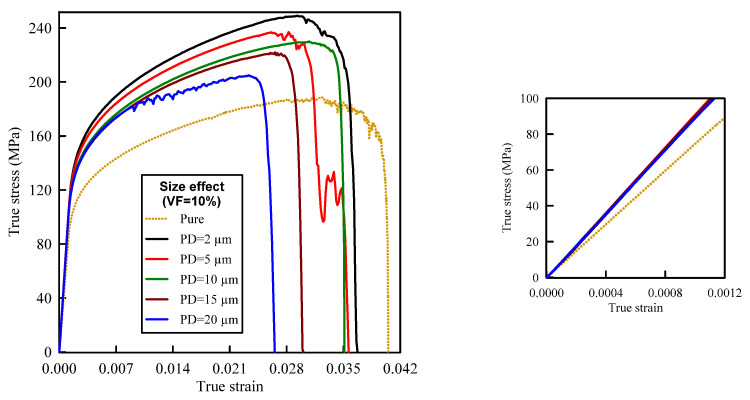
Effect of SiC particle size on the true stress–strain curve of A359/SiC composite (10% VF).

**Figure 14 materials-14-02143-f014:**
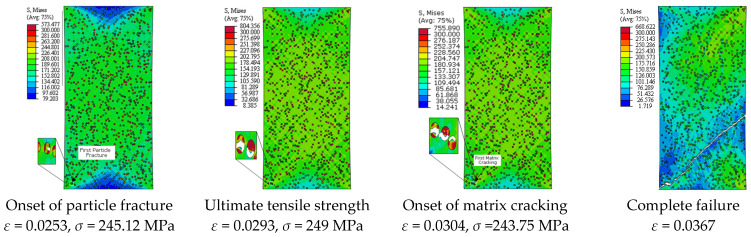
Failure propagation and von Mises stress distribution of A359/SiC composite at PS = 2 µm (10% VF).

**Figure 15 materials-14-02143-f015:**
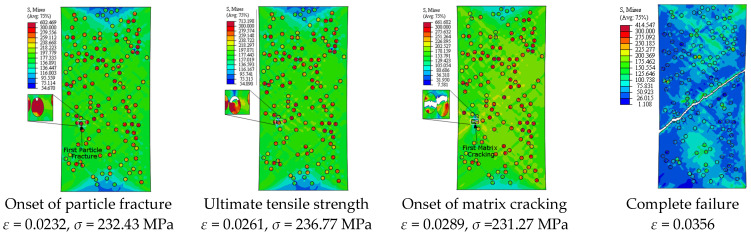
Failure propagation and von Mises stress distribution of A359/SiC composite at PS = 5 µm (10% VF).

**Figure 16 materials-14-02143-f016:**
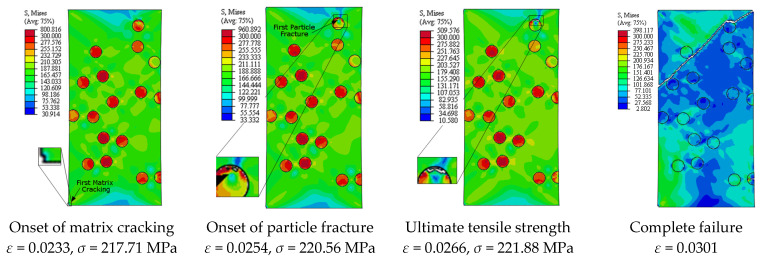
Failure propagation and von Mises stress distribution of A359/SiC composite at PS = 15 µm (10% VF).

**Figure 17 materials-14-02143-f017:**
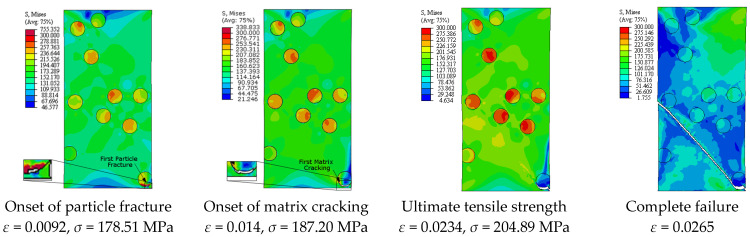
Failure propagation and von Mises stress distribution of A359/SiC composite at PS = 20 µm (10% VF).

**Figure 18 materials-14-02143-f018:**
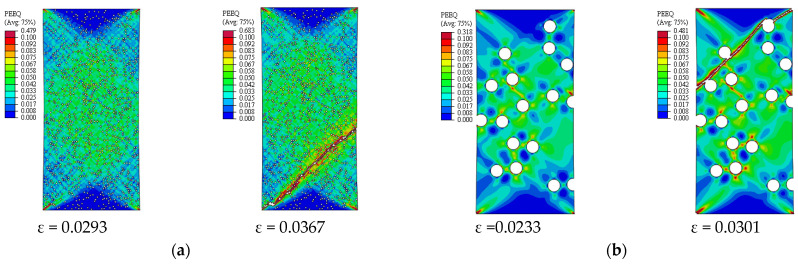
Effective plastic strain in the matrix of A359/SiC composite (10% VF): (**a**) PS = 2 µm; (**b**) PS = 15 µm.

**Figure 19 materials-14-02143-f019:**
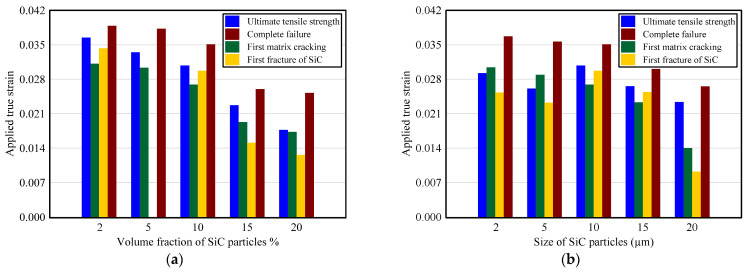
Variation of the applied true strain required for the onset of each failure mode and complete failure: (**a**) effect of VF; (**b**) effect of PS.

**Table 1 materials-14-02143-t001:** Mesh size effect on the failure strain and ultimate tensile strength.

Mesh Size (µm)	Element Number	Failure Strain	Ultimate Tensile Strength (MPa)
0.35	161,709	0.035072	235.5680
0.7	48,201	0.035117	230.0087
1.0	35,028	0.032361	218.6539
1.4	18,015	0.030625	208.4169

**Table 2 materials-14-02143-t002:** Effect of SiC particle volume fraction on the mechanical properties of A359/SiC composites (PS = 10 µm).

Property	VF = 0%	VF = 2%	VF = 5%	VF = 10%	VF = 15%	VF = 20%
Modulus of elasticity (GPa)	74.70	76.97	83.27	90.46	93.85	101.38
0.2% offset yield strength (MPa)	126.60	132.75	142.87	155.02	171.82	177.47
Ultimate tensile strength (MPa)	188.59	203.39	216.57	230.01	237.66	243.10
Failure strain	0.0405	0.0388	0.0383	0.0351	0.026	0.0252

**Table 3 materials-14-02143-t003:** Effect of the SiC particle size on the mechanical properties of A359/SiC composite (10% VF).

Property	Pure	PS = 2 µm	PS = 5 µm	PS = 10 µm	PS = 15 µm	PS = 20 µm
Modulus of elasticity (GPa)	74.70	89.27	90.13	90.46	89.99	89.12
0.2% offset yield strength (MPa)	126.60	167.68	162.68	155.05	153.61	153.35
Ultimate tensile strength (MPa)	188.59	249.00	236.94	230.01	221.88	204.89
Failure strain	0.0405	0.0367	0.0357	0.0351	0.0301	0.0265

## Data Availability

Data sharing is not applicable to this article.

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
