# Peer review of "Predictive Computational Model for Damage Behavior of Metal-Matrix Composites Emphasizing the Effect of Particle Size and Volume Fraction"

_materials, 2021, doi:10.3390/ma14092143_

Round 1

Reviewer 1 Report

Title: " Predictive computational model for damage behavior of metal- matrix composites emphasizing the effect of particle size and volume fraction"

In this work, the authors provided an integrated numerical model to investigate the effects of particulate size and volume fraction on the deformation, damage, and failure behaviors of particulate- reinforced metal matrix composites (PRMMCs). In the framework of a random microstructure- based finite element modelling, the authors modeled the plastic deformation and ductile cracking of the matrix by using Johnson-Cook constitutive relation and Johnson-Cook ductile fracture model. The authors simulated the matrix-particle interface decohesion through the surface based-cohesive zone method, while the particulate fracture was manipulated by the elastic-brittle cracking model, in which the damage evolution criterion depends on the fracture energy cracking criterion. In addition, the authors deveoped a bidimensional nonlinear finite element model by using ABAQUS/Explicit to model and analyze damage mechanisms of silicon carbide reinforced aluminum matrix composites. The authors claim that the predicted results have shown a good agreement with the experimental data in forms of true stress-strain curves and failure shape. Finally, a comprehensive parametric study was performed to investigated the influence of the volume fraction and size of SiC particles on the deformation, damage mechanism, failure consequences, and stress-strain curve of A359/SiC particulate composites.

General comment: This work should be revised to improve its quality and impact. In addition, it is not clear the value of this work with respect to the reference [20]. Again, the predicitve value of this model should be better explained with respect to literature or further data. The implementation of the distribution of the particle is not clear. Moreover, standard sections of a scientific manuscript should be implemented as "Introduction", "Materials and Methods", "Discussion", "Conclusion".

Some specific comments:

Lines :" Experimental investigation of damage and fracture behaviors in PRMMCs is very 41

difficult and requires high capabilities. To overcome this problem, computational FE 42

models of PRMMCs have been employed based on two structural-phenomenological lev- 43

els, i.e., macroscale, and microscale."

*) Could the authors explain in a better way why experimental results are difficult to achieve ?

Lines: "Recently, the damage mechanisms of 92

PRMMCs reinforced with different shapes of particles, individually or mutually, employ- 93

ing FE model based on random distribution of particles."

*) Please explain better the meaning of these lines.

Lines: "From the literature survey, it is evident that the influence of volume fraction and size 97

of particles on the damage behavior, failure mechanisms, deformation response, and ma- 98

terial properties of PRMMCs have not been sufficiently explored. This study aims to de- 99

velop an integrated 2D microstructure-based FE model in conjunction with image pro- 100

cessing and recognition for predicting the deformation and fracture of PRMMCs. The 101

model considers the random distribution of particles without varying its configurations. 102

The proposed integrated model accounts for the elastoplastic behavior and cracking of 103

matrix, decohesion at particle-matrix interface, and fracture of particles using Johnson- 104

Cook plasticity and damage models, cohesive zone surface method, and elastic-brittle 105

cracking model, respectively. ABAQUS/explicit FE software combined with Digimat FE 106

commercial code is utilized to perform simulations. The developed model is employed to 107

extensively explore the impacts of content and size of SiC reinforcing particles on the me- 108

chanical performance, and damage and fracture behaviors of A359/SiC composites. 10"

*) Could the authors better explain why it is evindent that "the influence of volume fraction and size

of particles on the damage behavior, failure mechanisms, deformation response, and material properties of PRMMCs have not been sufficiently explored." ?

*) Why " the model considers the random distribution of particles without varying its configurations". ?

Line : "he damaged stress s? is given 120"

*) What is the damaged stress ?

Lines :"The Macaulay bracket ⟨ ⟩ implies that damage onset is not affected by the compressive 138

normal stress, etc.. "+ Equation (6)"

*) This notation should be further clarified, since in Eq. (6) perhaps a different definition of <Tn> has been proposed with respect to the line 139

Equation(7)

*) Perhaps brackets are not balanced. Please correct.

3. The Finite Element Model

*) It is not clear the novelty of the proposed approach with respect to this one described ini "[20] S.I. Gad, M.A. Attia, M.A. Hassan, A.G. El-Shafei, A random microstructure-based model to study the effect of the shape of 414

reinforcement particles on the damage of elastoplastic particulate metal matrix composites, Ceram. Int. 47(3) (2021) 3444-3461. 415

https://doi.org/10.1016/j.ceramint.2020.09.189 41

"The mechanical properties of materials are greatly affected by the occurrence of the first damage during the loading process. This study presents a comprehensive investigation of the effect of particulate shape and combination of two shapes or more, as a design parameter, on the damage mechanism of particulate-reinforced metal matrix composites (PRMMCs). In the context of a random microstructure-based finite element modelling, the proposed model accounts for all possible failure modes; plastic deformation and ductile cracking of the matrix, matrix-particle interface decohesion, and brittle fracture of the reinforcement particles. The matrix plastic deformation and cracking are, respectively, modeled via Johnson-Cook constitutive relation and Johnson-Cook ductile fracture model. The cohesive zone method is adopted to simulate the particle-matrix interfacial debonding. The particle fracture is simulated using an elastic-brittle cracking model, in which the damage evolution criterion depends on the crack opening energy. An extensive parametric study is carried out to explore the effect of different shapes of SiC particles; circular, hexagons, squares, triangles, and their combinations, on the damage mechanism and consequences of failure of A359/SiC particulate composites."

*) Please explain in detail. Perhaps also in this work some further detail about the convergence issues, and the stability of the proposed algorithm should be proven for interested readers.

Section: "4. Results and Discussions"

*) This section should be reworked, since the Results should present the main results of the work without comments, while the "Discussion" should discuss the novelty of the proposed approaches with respect to the current state of the art. Please split in two sections...

*) The authors should explain in a detailed way whether this work is different from reference [20] and also the plots are different.

Section: "5. Conclusion"

*) The authors should provide a "Conclusion" section where the main achievements of this approach are clearly defined. A further a quick comparison to the current state of the art seem to be needed.

  • ) The nature of the random disposition of particles within the material is not clear. Please clarify.

Author Response

Please see the attachment. "Response to Reviewer 1 Comments"

Reviewer 2 Report

Dear Authors,

Thank you for the opportunity to review your work.

The paper is well written and covers an interesting topic. The process used in the study appears to be systematically and methodologically applied.

Similarly, Tables and figures are relevant however in some instance the titles are labelled are unclear for example correct the scale bars in Fig 10. Also, the font size on the images should be increased as they are difficult to read. Especially the legends. These images should be improved. 

Appropriate headings are provided and variables are clearly defined and measure appropriately. Finally, the details provided in the study are sufficient to replicate the study.   

Overall this is a good article that I would recommend for publication when these errors are corrected. 

Author Response

Please see the attachment "Response to Reviewer 2 Comments".

Reviewer 3 Report

In this paper, an integrated numerical model is proposed to investigate the effects of particulate size and volume fraction on the deformation, damage, and failure behaviors of particulate reinforced metal matrix composites (PRMMCs). In the framework of a random microstructure based finite element modelling, the plastic deformation and ductile cracking of the matrix are, respectively, modelled using Johnson-Cook constitutive relation and Johnson-Cook ductile fracture model. A two-dimensional nonlinear finite element model was established using ABAQUS/Explicit software, and the damage mechanism of silicon carbide reinforced aluminum matrix composites was modeled and analyzed. This article is innovative and can be accepted with minor modifications.

  1. The format of the article needs to be unified. Sometimes "Fig.X" and sometimes "Fig. 1" appear in the description of the Figure in the article. For example, "Figure 1" appears in line 177 and "Fig. 1" appears in line 189, and "Figures 4-8" appears in line 236, but "Fig.5" appears in line 246.
  2. There are some minor problems in the typesetting of the article.For example, the icon in line 218 of the article should be on the same page as the image, and so should the icon in line 250 of the article.Minor changes to the article are recommended.
  3. The explanation and explanation of the pictures in the article are relatively lacking, so it is necessary to make a specific explanation of the charts in the article. For example, for Figure 2 and Figure 3, the article does not specify the content of the picture. For example, "Figures 4-8" appears directly in line 236, and it is suggested to be modified into separate statements. There are also 288 lines "Figures 6 and 12-15".
  4. Why does ") "appear in line 248 of the article? It is suggested to remove") "and recheck the article to avoid such problems.
  5. The conclusion part is suggested to be expressed in points, which is relatively clear and the logic of the article is relatively strong.
  6. The conclusion part needs to be reorganized, and the innovation points in the article and my own contributions should be explained in detail. It is better to have some quantitative indicators.
  7. In this paper the effect of SiC particles volume fraction machine forehead SiC particle size has carried on the simulation research, respectively, and obtained the relevant law, but we still have some questions, the author besides the simulation to simulate the process, whether the related experiments, and the experimental results and simulation results were compared, thus proving the correctness of the simulation model is proposed by the author.

Author Response

Please see the attachment "Response to Reviewer 3 Comments"

Reviewer 4 Report

  1. In this paper “Predictive computational model for damage behavior of metal-matrix composites emphasizing the effect of particle size and volume fraction,” the effect of particulate size and volume fraction on the deformation, damage, and failure behaviors of particulate-reinforced metal matrix composites is investigated using finite element method.
  2. The reviewer cannot see any noticeable advancements in this paper from the authors’ previous work [20]. The theory applied is exactly identical. There is nothing new in the theory and numerical framework. The authors just solve slightly different applications (particle size and volume fraction change) from the previous work and based on the reviewer’s understanding, this cannot be accepted in the journal publications. The authors at least have to put more effort to make this manuscript look different from the previous work. They may focus on the development of numerical algorithms. The obtained numerical results are expectable according to the increase of the volume fraction and particle size.
  3. The reviewer cannot agree with the publication of this article in MDPI Materials. 

Author Response

Please see the attachment "Response to Reviewer 4 Comments"

Round 2

Reviewer 1 Report

Title: " Predictive computational model for damage behavior of metal- matrix composites emphasizing the effect of particle size and volume fraction"

In this work, authors provided the an integrated numerical model to investigate the effects of particulate size and volume fraction on the deformation, damage, and failure behaviors of particulate- reinforced metal matrix composites (PRMMCs). In the framework of a random microstructure- based finite element modelling, the authors modeled the plastic deformation and ductile cracking of the matrix by using Johnson-Cook constitutive relation and Johnson-Cook ductile fracture model. The authors simulated the matrix-particle interface decohesion through the surface based-cohesive zone method, while the particulate fracture was manipulated by the elastic-brittle cracking model, in which the damage evolution criterion depends on the fracture energy cracking criterion. In addition, the authors deveoped a bidimensional nonlinear finite element model by using ABAQUS/Explicit to model and analyze damage mechanisms of silicon carbide reinforced aluminum matrix composites. The authors claim that the predicted results have shown a good agreement with the experimental data in forms of true stress-strain curves and failure shape. Finally, a comprehensive parametric study was performed to investigated the influence of the volume fraction and size of SiC particles on the deformation, damage mechanism, failure consequences, and stress-strain curve of A359/SiC particulate composites.

General comment: The authors revised the manuscript. Although some scattered language errors  within the main text still make difficult to follow the logic flow in a totally clear way, the overall quality of the work has been improved. Please, fix all the minor linguistic issues, improve the quality of figures where needed (e.g., Figure 3 (a) has some labels inside the matrix area)  and improve captions where needed. The title of the "Discussion" sections should be changed accordingly (now the title is "Discussions"). Perhaps, a list of points within the "Conclusions" section is a suboptimal way to present the main achievements of this work....

Author Response

Please see the attachment "Response to Reviewer 1 Comments (Round 2)"

Reviewer 4 Report

1.    From the authors’ response below to the reviewer’s comments, the reviewer still cannot find any innovative or meaningful results or methodology enough for the publication. As the authors mentioned, the same FE model is utilized to solve a slightly different application.
In Reference [20], the effect of the particulate shape, considering different particle shapes and combinations of these shapes, on the damage behavior of PRMMCs was investigated using an integrated computational FE model. The model predicts the ductile cracking of elastoplastic matrix, decohesion at the matrix-particulate interface, and the brittle fracture of the reinforcing particulates. In the present paper, the same integrated computational FE model is employed to explore the effect of the volume fraction and size of the particulate on the damage behavior; sequences of failure, the required applied load for each failure mode, and the mechanical properties of A359/SiC composites, in addition to the distributions of the effective plastic strain are investigated.

2.    The strain gradient plasticity theory may be used to make a difference with [20] as was done in reference [10]. The results can be compared to those in [20]. In addition, there is one more reason that the strain gradient plasticity theory is needed. In Figure 2, a strong mesh dependency is resulted, which is physically and numerically unacceptable. It does not make sense to choose 0.7 µm just because it provides the closest results to the experiments only in this simulation. All the numerical results from different mesh sizes have to be identical and they have to be all in good agreement with the experiments. The authors may refer to Voyiadjis and Song (2019) in International Journal of Plasticity for a review of strain gradient plasticity with the effect of particle size on the mechanical behavior of PRMMC. 

3.    In lines 129 and 130, where do those values come from? Put the reference.

4.    In Figures 6-17, a unit of stress is missing.

Author Response

Response to Reviewer 4 Comments (Round 2)

Round 3

Reviewer 4 Report

Thanks for your time and effort to prepare the responses to the reviewer's comments and to improve the manuscript. The manuscript can be accepted in the current form.

This manuscript is a resubmission of an earlier submission. The following is a list of the peer review reports and author responses from that submission.